# Tumor Organoids as a Research Tool: How to Exploit Them

**DOI:** 10.3390/cells11213440

**Published:** 2022-10-31

**Authors:** Tijmen H. Booij, Chiara M. Cattaneo, Christian K. Hirt

**Affiliations:** 1NEXUS Personalized Health Technologies, ETH Zurich, 8093 Zurich, Switzerland; 2IFOM, FIRC Institute of Molecular Oncology, 20139 Milano, Italy; 3Beth Israel Deaconess Medical Center, Harvard Medical School, Boston, MA 02215, USA

**Keywords:** tumor organoid culture, personalized/precision medicine, tumor microenvironment, screening library, high-throughput screening, high-content screening

## Abstract

Organoid models allow for the study of key pathophysiological processes such as cancer biology in vitro. They offer insights into all aspects covering tumor development, progression and response to the treatment of tissue obtained from individual patients. Tumor organoids are therefore not only a better tumor model than classical monolayer cell cultures but can be used as personalized avatars for translational studies. In this review, we discuss recent developments in using organoid models for cancer research and what kinds of advanced models, testing procedures and readouts can be considered.

## 1. Introduction

Cancer research is and has been an important field in biomedical research. It initially started with mostly observational studies linking exposure to increased cancer risk [1]. With the advancement of pathohistological technologies, it has been possible to study patient-derived tissues and follow-up changes on the cellular levels. In recent decades, the advent of molecular research further advanced the field as well as highlighted potential vulnerabilities. For example, targeted treatments have shown a promising response rate using preclinical models [2]. However, the subsequent clinical studies failed to show the same high correlation between molecular targets and clinical responses [3], emphasizing the need for better preclinical and translational models.

Traditionally, cancer cell lines have been the most commonly used preclinical model. Unfortunately, only a minority of highly aggressive tumor cells are able to be cultured classically in vitro as a monolayer [4,5]. Those cancer cell lines are able, as model systems, to recapitulate pathophysiological changes but are limited in their translational potential. Organoid cultures, on the other hand, allow for the expansion of cells that would otherwise not proliferate in vitro. They are able to maintain in vivo phenotypes such as complex organization, tissue-specific functions, genetic stability and specific disease-state phenotypes.

In the context of this review, we consider organoid cultures as primary, in vitro and 3D cultures which have the ability to retain heterogeneity and expansion potential. Organoid cultures allow for a long-term culture of adult stem cells with a high efficiency [6]. Some adaptations to the initial protocol for gastrointestinal tissue have been proposed, but, classically, cells or clusters of cells are embedded within an extracellular matrix as a 3D substrate, and a medium containing a cocktail of defined growth factor is used throughout the culture period. The protocol has been adapted to allow for the culture of the major epithelial cells derived from either murine or human tissue. Culture protocols exist for the colon, liver, pancreas, lung, bladder, prostate, ovaries, esophagus, endometrium, brain and more tissue types [7,8]. This is in contrast to other 3D cultures, such as spheroid or tumoroid cultures, where classical immortalized monolayer cell lines are used to generate 3D tissue.

It has been recognized that primary organoid cultures, especially from cancer patients, are a valuable resource, and, therefore, efforts have been generated to establish academic or commercially available biobanks for organoids [9,10,11,12,13,14]. Many prior reviews on organoids emphasize basic and translational research questions that organoids can answer [7,15,16,17,18]. This review focuses on organoids derived from adult stem cells in the context of cancer research. We are aiming to spotlight advances in screening technologies, guided by why, when and especially how to exploit them.

## 2. Advanced Models

### 2.1. Improved Substrates and Cultures

In order to develop three-dimensional cell culture models such as tumor organoids, the cells need to be provided with an environment that allows them to develop into a three-dimensional structure. The properties of the culturing environment can have significant implications for the assays’ physiological translatability as well as for the automation of the protocol. 

The vast majority of organoid assays use an animal-derived extracellular matrix extract most commonly known as Matrigel (other commercially available alternatives exist). The use of Matrigel for organoid cultures is well described, and protocols can be found with ease [19]. Matrigel is known to be biologically active and induce cell differentiation [20]. Matrigel is an extract of the Engelbreth–Holm–Swarm mouse sarcoma that contains large quantities of basement membrane components such as laminin, collagen IV, perlecan and others, including growth factors. Variations in the production process have been implemented to develop certain variants with reduced levels of growth factors or increased levels of collagen [20]. Gelation is induced by warming the material to temperatures between 24 and 37°, but cooling does not re-dissolve the gel. However, the composition of Matrigel, being a natural product, is not well defined and is sensitive to batch variations [16]. Additionally, the gelation properties of Matrigel make it less suitable for large-scale experiments, as the temperature requirements impose limitations on handling the gel with automation equipment. To improve standardization, it is also possible to use a single material such as collagen in a pure form [21,22], but both collagen and Matrigel suffer from temperature-dependent polymerization that makes them less ideal for lab automation purposes. Collagen or Matrigel can also be used in combination with other materials, such as the combination of collagen and nanocellulose, which can greatly reduce the number of reagents required and also allow for faster polymerization [23]. Alternatively, one can use nanofibrillar cellulose [24] or alginate gels [25] or use synthetic polymers or recombinant peptides [26,27]. Synthetic hydrogels, as well as nanocellulose or alginate gels, have the advantage that their properties can be much more easily controlled and better defined than natural hydrogels such as Matrigel and collagen, placing them at a clear advantage for automated experiments and screens. However, these gels are often not biologically active, which, in turn, lowers the physiological relevance of the 3D culture environment [26,27,28,29,30,31]. A final consideration is to not use a scaffold at all: certain organoid types can form in suspension cultures using ultra-low-attachment (ULA)-treated plates [32], microcavity arrays [33] or microwell plates [34]. Due to the elimination of the matrix requirement, such approaches are readily scalable, although the plate format can put restrictions on the degree of assay miniaturization.

### 2.2. Complex Models

Model systems are the workhorse of cancer research, allowing for rapid hypothesis testing using a limited complexity system. The drawback of those limitations is the translational oncology dilemma, where many compounds later fail to demonstrate efficacy during in vivo validation studies or early clinical trials [35]. Advanced in vitro models are therefore incorporated in preclinical research. The basic components of any cancer model are the cell source and subsequent culture conditions, which would be preferentially close to the tumor growth conditions observed in a patient. Even though it is technically possible to keep the tumor microenvironment intact using tumor fragment culture, the expansion potential, culture time and reproducibility, which are necessary for high-throughput studies, remain limited. 

In this section, we want to provide an overview of advanced organoid cultures, such as more complex organoid models which incorporate additional components of the tumor microenvironment (TME), genetically engineered organoid cultures and personalized organoid models. One possible way to study tumor cells in the context of their own TME has been proposed with the advent of Air–Liquid Interface (ALI) culture [36,37]. In this method, tumor cells are cultured as organoids en bloc, alongside native stromal and immune components. The great advantage of this technology relies on the preservation of the native TME, allowing for the study of tumor cells growth and interactions in its own structural environment [36,37,38]. However, the preservation of this complex system over time has proven difficult compared to more “traditional” organoid cultures, both in terms of tumoral architecture and immune cell survival and functionality [36]. 

#### 2.2.1. Organoid to Study the Interaction with the Immune System

The advent of cancer immunotherapies has revolutionized the treatment of advanced cancers: not only have checkpoint inhibitors been successfully used for the treatment of multiple tumors [39], but the adoptive transfer of autologous tumor-infiltrating lymphocytes has also shown impressive clinical results [40]. However, although long-lasting clinical responses have been described, the response rates to current immunotherapies remain modest overall across the different tumor types [41,42]. To exploit the full potential of immunotherapy, it is necessary to find ways to predict which patient is most likely to respond or not to the treatment, who is likely to eventually relapse and, most importantly, how to overcome the rise of resistance mechanisms. These challenges highlight the need for a pre-clinical model that allows for the interrogation of the interaction between tumor cells and immune cells, in an unbiased manner, for the individual patient. Traditionally, cancer research utilized 2D cell cultures and in vivo xenografts or genetically engineered animal models. However, these models insufficiently capture the complexity of the immunobiology of native human tumors [38]. A paradigm change happened with the emergence of cancer organoids pure 3D epithelial cell culture, which is able to recapitulate the histological and molecular features of the tumor of origin [43,44]. By co-culturing tumor organoids with native or reconstituted autologous TME immune components, it has been possible to dissect the interaction between tumor cells and immune cells for the individual patient (Figure 1). For example, Tsai et al. describe how to co-culture human pancreatic cancer organoids together with autologous CAFs and T-cells [45]. This method has not only been proven suitable for the precision testing of drug responses but also for testing the lymphocytes infiltrating capacity into the tumor tissues of these models. A coculture of pancreatic organoids and CAFs has also been used by Biffi et al. for the identification of tumor-secreted ligands capable of promoting inflammatory CAF subtypes [46]. In addition, a coculture of mouse-derived primary 3D mammary epithelial organoids with autologous CD4+ T cells and TAMs was proven successful for the study of the invasiveness of mammary carcinomas [47]. Moreover, an interesting method to investigate the toxicity of CAR-NK cells has been described by Schnalzger et al. A pre-clinical platform to identify and select suitable target antigens for CAR therapy was developed by comparing the cytotoxic effect of CAR-NK cells against matching tumoral and normal colon organoids [48]. 

All of the previously described methods focus on the interaction between tumor organoids and a single immune population of interest. A different path has been taken by Dijkstra et al. Here, patient-derived tumor organoids are cocultured in the presence of autologous circulating peripheral blood lymphocytes, without any selection of a particular subset of the immune population. This study provided a proof-of-concept for a new unbiased strategy for the generation of tumor-reactive cytotoxic T cells for patients with MSI-colorectal cancer or non-small cell lung cancer [44,49].

#### 2.2.2. Dynamic Organoid Cultures Incorporating Vascularization and Fluidic Technologies

A major limitation in current standard organoid cultures is their reduced tissue maturation and constraints on tissue size, both of which have been, in part, proposed to be related to the lack of a functional vasculature [50]. Several approaches have been proposed to help vascularize organoids using a co-culture with endothelial cells, co-differentiation with progenitor cells and/or mechanical stimulation. Interestingly, these mostly focus on benign and not cancer organoid cultures.

The co-culture approach where endothelial cells are mixed with epithelial cells at a defined ratio seems to be the most intuitive way to vascularize organoids. Successful generation relies on a defined plan in regard to the timing, ratio and organoid protocols. This approach has been used in the generation of vascularized liver and pancreatic organoids [51,52,53]. Co-differentiation with mesodermal progenitors more closely mimics the process of organogenesis; however, it cannot easily be controlled. Interestingly, this approach has been successful in the generation of advanced kidney models which integrate nephrons, collecting duct networks, renal interstitium and endothelial cells [54]. By using the forced overexpression of transcription factors in human-induced pluripotent stem cells, vascularized and patterned neuronal organoids could be formed [55]. A protocol for complete blood vessel organoids has been developed, which exhibits the morphological, functional and molecular features of a microvasculature [56].

On the other hand, fluidic technologies such as mechanical stimulation allow for an improved differentiation. For the generation of brain organoids, spinner flasks or smaller bioreactor systems are part of the protocol [57,58]. If larger numbers of organoids are needed, either as a cell source or for a larger screening project, using a spinner flask culture can increase the expansion up to 40-fold compared to a 6-fold expansion in static cultures [59]. On the more microfluidic site, studies have shown that chip platforms allow for dynamical screens with variations in the concentration, timing and duration of fluidic drug delivery [50]. Such approaches facilitate combinatorial screens and temporal administration better mimicking the clinical situation.

#### 2.2.3. Genetically Engineered Organoid Models

Organoid cultures are amenable to genetic modification, and, with the advancement of the CRISPR/CAS9 toolbox, they have been successfully used as modified models in several studies. The efficiency to genetically modify primary cultures is generally lower than classical monolayer cell cultures but can be increased using a positive or negative selection. Initially, genetical studies with organoids have been performed to generate models of tumor progression, as best described for colorectal cancer [60,61]—more recently, this has been implemented in the functional validation of less frequently observed driver genes [62,63]. Genetic screening allows for an unbiased evaluation of gene function using a pool of barcoded vectors. More recently, this technology has been applied to screen with primary tumor organoid cells instead of using cancer cell lines [64,65], which helped to evaluate the resistance to key growth factors related to colorectal cancer progression. The combination of functional screens with genetic modification is especially important to validate drug–gene interactions. In our own study, previously described vulnerabilities in common mutated genes were evaluated using unmodified and genetically engineered pancreatic cancer organoid lines [66]. As observed in clinical studies, we could show that a single mutation may not be enough to predict a response, and other factors such as tissue type and mutational signatures may be important additional factors to take into consideration.

#### 2.2.4. Personalized Models & Biobanking

Cancer organoids have been proposed as a model system for precision medicine. Indeed, their capacity to retain the characteristics of the original tumor makes them a unique model for cancer research in the personalized setting [7]. In 2015, it was demonstrated that organoids reflected the tissue from which they were derived. Comparative genetic analysis between tumor organoids and the tissue of origin in 14 different patients affected by metastatic colorectal cancer showed that 90% of somatic mutations were shared; also, a correlation of nearly 0.9 was observed at the level of DNA copy number profiles [43]. Moreover, tumor organoids morphologically reflect the original tumor from which they are derived. A side-by-side comparison of H&E staining for both colorectal and non-small cell lung patients clearly showed the similarity in terms of architecture and 3D structure between organoids and the tissue of origin [44]. In addition, an attractive feature of organoid models is the wide range of epithelial tissues from which they can be derived; therefore, they are successfully expanded in the long term to reach large living biobanks. Given the relative ease of organoid culturing and expansion to a large number and the stability of their morphological and genetic features over passages, large living biobanks for different tumor types have been generated [8,9]. Altogether, these remarkable features of the organoid culture model make them extremely attractive in studying personalized anticancer treatment. However, many challenges need to be resolved to fully exploit the potential of tumor organoids for clinical decision making. The success rate of establishment can vary greatly among tumor types, spanning from ~70–90% for colon tumors to as little as ~20% for lung tumors [8,9], limiting the clinical translation to a smaller fraction of patients. Furthermore, once established, the growth rate of a culture differs among patient samples, as well as tumor types [7]. For example, it could take from 25 to 170 days to generate an NSCLC organoid line [49], highlighting how, for some slow-growing patient-derived samples, the timeline of organoids establishment is incompatible with the clinical decision process.

On the more clinical side, with the emerging evidence indicating that organoids are a helpful pre-clinical model, the interest arises to study them in a more translational setting. In the study led by Vlachogiannis et al., patient-derived organoid models were shown to have a high sensitivity and specificity in forecasting the response to targeted agents or chemotherapy in patients [67]. Following this finding, multiple oncological studies have been performed, which have been systematically reviewed [68]. The pooled sensitivity was 0.81 and the specificity was 0.74 for using organoids to discriminate patients with a clinical response. As the authors mentioned, many of those have been performed on a more heterogeneous group of cancers and treatments. Additionally, the readout was different between studies. Clinical studies using similar and larger patient cohorts are necessary, especially those addressing cost-effectiveness and the clinical benefit for patients.

## 3. Advanced Application

Repurposing studies with smaller tailored drug libraries as well as (ultra-) high-throughput screens with large, diverse chemical libraries are frequently used strategies to identify new lead candidates in the field of oncology [69,70]. Using patient-derived tumor organoids to test drugs holds great promise in identifying the best personalized therapy for a patient and has the potential to lead to the discovery of novel lead candidates that can benefit other patients [71] (Figure 2). Table 1 includes an overview of small- to large-scale screens that have been published using patient-derived tumor organoid models for different diseases. Although this is not an exhaustive list of all screens with organoids, it is clear that many drug evaluation approaches use standard-of-care treatments or related chemotherapeutic drugs at varying concentrations to profile responses, rather than testing large panels of untested compounds.

### 3.1. High-Throughput Drug Screening

The preparation of organoids from patient tumors (either from tumor biopsies or from surgically resected tissue) for drug evaluation can be carried out on a small scale and can help guide the clinical decision-making process [72,73,74,75,76,77] (Table 1). The preparation of organoids from patient tissue is a laborious process, and as a result of high costs and often limited tissue material, this approach is not yet undertaken systematically for cancer patients. A challenge, among others, is that some tumor types have a relatively low success rate for developing organoids [78,79], and the time needed to generate and expand organoids for drug evaluation might, in some cases conflict with the timeline of a therapeutic regimen proposal. In cases where it is possible to develop organoids and use them for drug evaluation in a timely manner—for example, within the patient’s recovery period after surgery—there are large potential benefits to patient survival [74,80,81,82].

Even though tumor organoids have much a larger potential to directly benefit the patients compared to most other current in vitro technologies, it is still important to realize that a tumor organoid is a disease model, meaning that several critical components may be missing that are needed to really capture the in vivo situation: circulation/vasculature, the immune system and the tumor extracellular matrix. Although, as we discussed above, strategies are emerging to allow for a more faithful representation of the tumor tissue. Considering that the particular in vitro growth conditions may affect the selection of the therapeutic candidates, the culturing of tumor organoids in vitro can introduce unwanted artificial influences such as residual growth factors present in culture media and/or ECM and changes in the pH level of the culture medium [94]. In addition, cell culture incubators are not necessarily providing a correct testing environment, as the oxygen gradients are usually not very representative of those in the human body [95]. Furthermore, the addition of test drugs to the culture medium for the evaluation of tumor organoids in vitro might not reflect the route of drug exposure when the drug is given to a patient; this could potentially alter the drug uptake by the cells and lead to a different response in vitro compared to the patient. An example of a situation where the exposure route might not reflect the in vivo route is in assays for cystogenesis [96]. Similarly, it is challenging to extrapolate the test concentration in vitro to an effective dosage for the patient, because the formulation, metabolism and duration of action of the drug can be very different. Nevertheless, it is interesting to link such data with advanced profiling services (e.g., Tumor Profiler) to support the decision-making process [97], and despite these differences, there seems to be a good correlation between in vitro, in vivo [89] and patient drug responses [68,88,98,99].

A particular niche where traditional in vitro models are often missing and where tumor organoids can really stand out is rare cancer types [100,101]. For example, they can be used to elucidate disease mechanisms of rare subtypes of prostate cancer [91] and can benefit patients with cholangiocarcinoma [102], appendiceal cancer [103,104], adenomyoepithelioma [105] and pancreatic neuroendocrine cancer [106].

The next sections focus on large-scale drug screening and lab automation; when purely considering applications for personalized therapeutics, automation in this context is less relevant due to the small selection of drugs that are typically evaluated [Table 1] and the high relative costs of the automation equipment. However, when such diagnostics approaches are undertaken routinely, automation equipment can improve the assay quality and performance through standardization.

#### 3.1.1. Considerations for Library Selection and Compound Management

As illustrated in Table 1, it seems that most screening efforts using tumor organoids use small collections of (typically) FDA-approved drugs for multi-dose evaluation, generally in (either technical or biological) duplicate or triplicate, but other types of libraries are less popular for screening on patient-derived material. This is perhaps related to the challenge of obtaining a sufficient number of organoids for high-throughput screening, since transformed/genetically engineered organoid lines have been used for larger-scale screening [107]. On the other hand, since compound collections that do not contain approved drugs are less likely to directly benefit the patient, there might be less of a need to screen larger collections of compounds using direct tumor organoids, since high-throughput screens usually focus on the identification of novel therapeutic entities rather than on providing patient-specific therapeutic recommendations. High-throughput screens, therefore, also have different requirements for the design of the compound libraries. Compound collections are often synthesized in-house or can be obtained from chemical vendors (e.g., SelleckChem, Enamine, ChemBridge, Asinex, Specs), and they can vary in size and composition a lot. On the one hand, it is typically possible to purchase smaller, more focused collections of compounds, such as, for example, collections of clinical compounds or FDA-approved drugs. On the other hand, it is possible to compile and purchase larger, chemically diverse collections that can additionally be filtered to have desirable drug- or lead-like properties. The correct selection of the library is likely to depend on the research question and on the screening setup: screens of smaller tailored libraries are attractive when a specific molecular target is known or when the assay throughput is limited, which is an important consideration for organoid screens. In target-agnostic screens with simple readouts that are more amenable for automation, it is therefore more attractive to screen large compound collections. Conversely, the selection of a compound collection also affects the screening setup: whereas a collection of approved drugs and clinical candidates are molecules already developed for therapeutic applications, collections of diverse chemical structures are not, affecting typical concentration ranges for testing. It is challenging to give particular recommendations, but approved drug collections often (depending on the tumor type and the individual patient) already show inhibitory activity at test concentrations up to 1μM [9,66,81,89], and screening at higher concentrations might lead to an excessively high hit-rate. Chemically diverse compound collections are generally less likely to provide a high hit rate, and the binding affinity of these compounds to certain receptors, kinases or other druggable targets is unlikely to be high. In this case, it can be useful to screen at higher concentrations, as long as the solvent (typically DMSO) concentration and compound solubility allow this [108]. The proper selection of the solvent for the compounds is important to avoid toxicity, chemical interference [109,110], solubility/stability [111,112,113] and evaporation. This can potentially lead to problems in libraries containing multiple solvent types.

#### 3.1.2. Automating High-Throughput Screens with Tumor Organoids

A key process improving assay reproducibility and, in general, being able to screen larger compound libraries is the assay miniaturization and automation. For assay miniaturization and automation, screens with organoids follow roughly the same optimization process as for any other cell type, although there might be larger challenges depending on the model. To successfully automate a screening assay, the first step is generally the assay optimization. This can include the miniaturization of the assay format from, e.g., a 96-well plate format to a 384-well plate format, but it always includes the generation and testing of methods generated on the automation equipment. A common misconception regarding assay miniaturization with cell-based assays is that the volumes and cell density can simply be scaled down to the new format. However, especially with organoid assays, miniaturization typically requires further optimization and extensive testing [83,107]. Core to this phase is also the comparison of the performance of the automated screening setup to that of the manually performed assay. During assay optimization, the correct plate type is also generally selected, but due to the many specialized plate formats that are now available commercially (e.g., ULA-treated plates [32], microcavity plates [33] or microwell plates [34]), it is important to evaluate if the plates follow SBS guidelines. In general, a good recommendation for readers intending to miniaturize and automate their screening setup is the Assay Guidance Manual [114]. As was already mentioned previously, a challenge in screening with tumor organoids is the duration of the organoid expansion process. With traditional screening setups, large excess volumes are required, and this is not ideal when materials are scarce or expensive (e.g., typically 5–10 mL of cell suspension is needed to fill the tubing of peristaltic pump cassettes, and about 10 mL of hydrogels may be additionally required to fill reservoirs for automated pipetting). The required amount of material can be minimized by miniaturizing the screening assay to either 384- or 1536-well plates, but the reduction of dead volumes requires the acquisition of specialized equipment.

There are several other challenges in the automation of organoid screens. The first and foremost limitation is the use of animal-derived hydrogels (which, as Table 1 illustrates, is still a very popular matrix for organoid cultures), as these require extensive temperature control. While it is not impossible to work with these gels using automation equipment, the temperature restraints might challenge the already existing screening infrastructure, and the typical requirement of pipette tips for transferring the gel make the assay expensive due to the materials and excess volumes required to fill reservoirs. The dispensing of these materials without pipette tips by using dispensers with, for example, a peristaltic pump can lead to tubes blocking due to the premature polymerization of the gels, and it typically requires cooling the entire equipment.

Secondly, due to the often prolonged assay time for organoids, there is a very real chance of edge effects on plates as a result of evaporation [115]. This is not an exclusive limitation to the automation of organoid screens but also happens when performing these assays manually. In the automated context, however, one ideally wants to also use the outer plate wells in order to screen more cost-effectively. Certain solutions to the problem exist—for example, gas-permeable plate seals or active incubator humidifiers can counteract the problem to a certain extent.

In order to enhance the possible throughput of the screening pipeline, it is important to identify the bottlenecks in the procedure. In most of the cases, this is either the drug dilution and addition (in case pipette tips are used) or the readout, possibly including washing steps (more about this in Section 4). Specialized scheduling software provided by lab automation integration companies can generally help to identify bottlenecks and allow for the optimal scheduling of the experiment. In addition, this software can also help to track samples in the system, which is relevant when processing many samples or plates. A very interesting technology in this respect is acoustic dispensers: they allow for pipette-tip-free liquid dispensing while also being able to track the sample condition and volumes, and they also give warnings if a liquid transfer fails, which is an important quality control parameter not otherwise obtained [116].

A final challenge with laboratory automation in general is the costs associated with the acquisition of the equipment and also with the maintenance of the equipment. Since specialized engineers often have to travel from abroad for servicing the equipment, maintenance contracts can be excessively expensive and especially challenging for academic institutions. The automation facilities additionally require skilled personnel to operate the equipment, and, considering that lab automation is a relatively new field in life science, it is challenging to find skilled personnel. It is also not uncommon that the training of personnel on an integrated lab automation solution takes between 6 months and a year. However, it is important to note that when automation equipment is properly used and set up, it can actually reduce the screening costs [117].

#### 3.1.3. Sources of Assay Variation: Factors to Consider for a Successful HTS Assay

Although not the main topic of this article, the quality of high-throughput screens is typically assessed by calculating a z’-factor. This is a statistical parameter with a maximum value of 1. Screens are typically considered to be successful if this z’-factor is 0.5 or higher [118], as this would indicate a sufficient separation between positive and negative control conditions. However, there is some debate about the exclusive usage of z’-factors to assess screen quality [119]. For some assay setups—especially more challenging setups such as those with organoids and more complex readouts—it is typically a problem to achieve z’-factors of 0.5 or higher. This does not necessarily mean that those assays are not useful, considering that assays with a z’-factor lower than 0.5 can also identify useful hits [120]. However, in order to provide screening data of good quality that are also reproducible, it is important to minimize sources of variation. While readouts will be discussed later on, the selected readout can have a large effect on the z’-factor: image-based screens typically have a lower z’-factor than whole-well measurements such as viability assays.

We already discussed the variation from batch-to-batch differences of reagents, but batch-to-batch variation can also be present in plastic consumables. For example, we have observed variation in plate- and pipette tip performance. This might be caused by variations in the type of plastic used in the materials but is also sometimes a result of improper surface treatments (e.g., TC-treated or ULA plates). We therefore recommend bulk purchases and the testing of each batch of reagent and labware prior to performing a screen.

An additional source of variation can arise as a result of increased numbers of liquid handling steps when working with organoids. Due to the prolonged duration of organoid assays, it may be required to exchange culture media and re-dose compounds. While this can be especially challenging for suspension cultures, such intermediate steps introduce further variation to the assay, which negatively affects the z’-factor.

A frequent issue with screening data is low reproducibility between different runs or laboratories. While this has been reported for drug tests with 2D cancer cell lines [121,122,123], it is unclear if this problem also affects screens with tumor organoids. A challenge is that the tumor organoids are patient-specific, and it is therefore difficult for other facilities or labs to completely replicate the assay conditions. To improve this, a more open exchange of biobanked patient-derived material and more detailed methodology descriptions are essential. As mentioned earlier, some concise methodology papers can be found for developing organoid-based screens, but it is often challenging to determine how certain drug dilutions or-exposures have been performed, especially in the context of automated screening, considering custom definitions of liquid classes and dispensing height settings. Additionally, it is often challenging to find crucial information such as at which density the cells or organoids were seeded into plates, at which concentration the drugs were screened or whether the included replications are biological or technical. When dispensing small volumes, such as those typically required to dispense into 384- or 1536-well plates, the relative humidity and temperature in a particular laboratory might influence the evaporation rate, thereby lowering the accuracy of dispensing steps and potentially also aggravating edge effects in plates. A practical solution to these problems would be to provide device-specific information and protocols as an appendix to publications of screening data and encourage authors to describe their methodology more extensively.

## 4. Advanced Readout

Organoids are complex microtissues with a large heterogeneity. To provide the most valuable data, this heterogeneity should ideally be reflected in the measurement of the results. However, as can be seen in Table 1, most drug evaluations and screens performed with organoids have cell viability measurements as the prime readout. While the reduction of cell viability might be an appropriate readout when evaluating therapies against tumor organoids, this type of measurement restricts the amount of information that can be extracted from the organoids, as the treatment response is generalized into a full-well measurement, which does not account for differences in cell populations within the organoid or between organoids if multiple organoids per well are present. Because cell viability assays such as CellTiter-Glo^®^(-3D, Promega) or resazurin are robust assays [124,125] that generally give a stable signal, these assays are popular choices for large-scale screens with large libraries of compounds, especially if no drug target is known (after all, anti-cancer drugs are typically expected to kill or at least impair the growth rate of tumor cells). A strategy here could be to identify viability-reducing hits using simple assays and subsequently validate the outcome using more advanced readouts such as multiplexed assays or image-based evaluation. One risk here is that drug efficacy on important cell populations within the organoid might be overlooked, as the effects could be subtle when considering the entire well as a whole. Additionally, in the case of co-culture approaches, these assays do not allow for discrimination between different cell types, and microscopy-based readouts are more suitable.

There are possibilities for multiplexing several readouts, which can improve the value of the information extracted from organoids. For example, it is possible to combine cell viability measurements with, for example, metabolic measurements or with imaging [126]. Important for such approaches is that the assay needs to be set up for both readouts, which puts restrictions on the type of plate one can use in screens. Conversely, certain readouts that require lysis of the organoids (e.g., CellTiter-Glo^®^3D) are only compatible with imaging if this measurement is performed last. However, live-cell imaging with organoids before an endpoint-readout in a screening context can be time-consuming, thereby limiting the possible size of the screen, as imaging-based readouts are often the bottlenecks of screens. A much more feasible approach to image-based screening is the fixation/staining of the plates and imaging the plates after the screen is completed. The optimal setup will of course depend on the research question: is single-cell resolution really necessary, or will a single vertical series of xy-images (z-stack) per well at a low magnification suffice to identify compound efficacy? In addition to often being time-consuming, image-based readouts for 3D cell culture models are also very data-intensive due to the generation of z-sections, often in multiple image channels. The already long imaging time makes high-magnification objectives less suitable for large-scale screens. One solution to the large volume of data generated is to generate image projections. While these are less data-intensive, they also cause loss of detail, and when multiple objects are present in one well, they might overlap vertically, appearing to be the same object in a projection. For more information on the optimization of imaging and phenotypic analysis in the context of 3D cell models, the authors refer elsewhere [127].

## 5. Conclusions and Future Perspectives

Organoids have been shown to be excellent models in answering a variety of scientific questions. Their major advantage is in the ability to grow primary tissue for an extended time, which allows them to generate personalized models. This is especially important in cancer research, as tumors are heterogeneous, and not a single model can account for all the phenotypes observed.

So far, this has been challenging due to the technical limitations and additional costs generated by these more complex models. Therefore, as of now, there have been no drugs approved using screenings with organoid technology. Many disregarded the utility of advanced cell-based models in drug discovery [35]. With the simplification of protocols and high-throughput availability, promising drug candidates are now identified [84].

The approach is key to success, and important questions remain at the start of any translational project. Some important general questions are as follows: Which organoid models best represent a disease? Do you need to add additional cell types to your model? How many are needed to be representative? Which screening platform and methodology is efficient and relevant? Which library is going to be used? What readouts are going to be considered?

The remaining challenges for organoid cultures are in regard to the standardization of culture conditions, the costs for substrates and growth factors and the difficulties in recapitulating the full (tumor) microenvironment. Organoid cultures are an important addition to the toolbox of cancer research and will hopefully contribute to increasing the translational potential of compounds identified preclinically.

We envisage a process where tumor organoids are able to be established in the future within a shorter timeline and with a higher efficiency by adapting available protocols. The standardization of assay procedures and the handling of materials would make the technology more available and more amenable for high-throughput drug screening. A move away from simple cell-viability measurements as assay readouts for drug screens, as well as the inclusion of multiplexed- or imaging-based readouts, will open the doors to identifying cell (-type)-specific effects of drugs, likely contributing to the development of drugs with fewer side effects and lower chances for developing resistance. Personalized models with human-derived or recombinant materials are more standardized and lead to focused clinical trials. More complex tumor-immune models should be used to explore pharmacological points of actions. Organoid models of disease may become even more important after the passing of the FDA modernization act 2.0 S.5002, which allows for the replacement of certain animal studies by using alternative models such as organoids. We expect that the passage of this act will cause a surge in the popularity of advanced cell-based models such as organoids, replacing certain animal models, which can also lead to an improved standardization of and certification of methodology.

## Figures and Tables

**Figure 1 cells-11-03440-f001:**
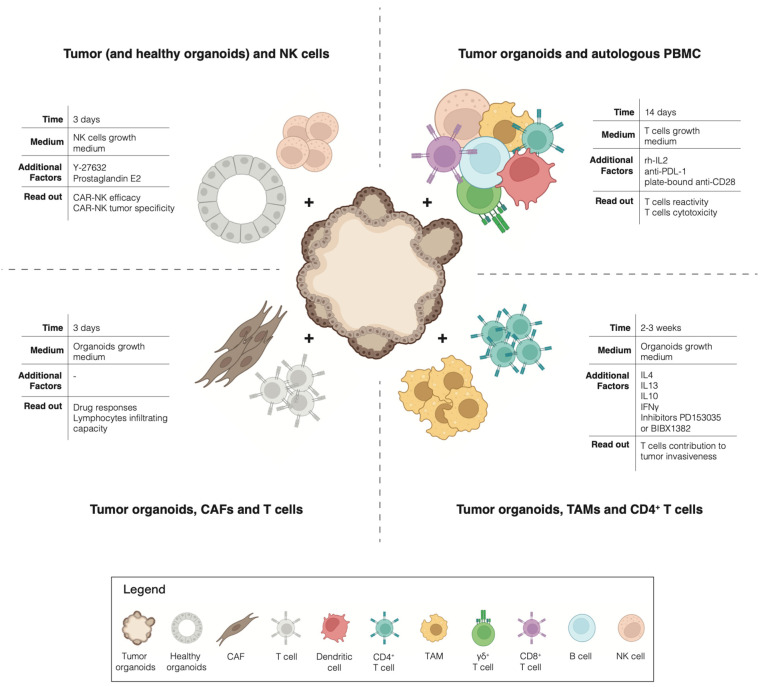
Overview of co-cultures.

**Figure 2 cells-11-03440-f002:**
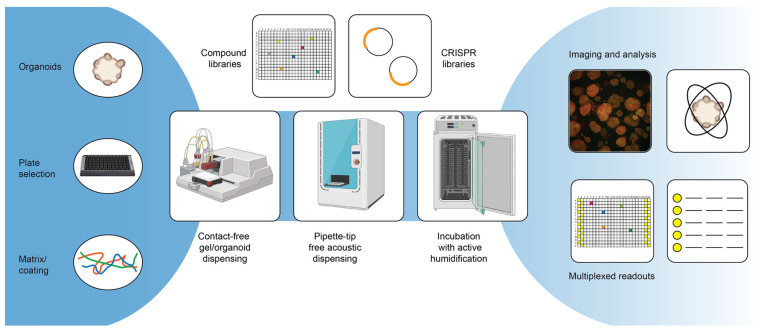
Advanced screening technologies.

**Table 1 cells-11-03440-t001:** Drug response tests and screens with human tumor organoids. ULA, ultra-low-attachment surface; FDA, federal drug administration; BME, (Cultrex) basement membrane extract; TC, tissue-culture. Coloring per type: Gastrointestinal, Genitourinary, Other.

Organoid Type	Screen Size	Plate Format	Matrix/ Plate	Library Type	Readout	Ref.
Colorectal cancer	83 compounds on 19 organoid types in triplicate	384	BME	Mixed	CellTiterGlo (viability)	[9]
Colon cancer organoids	-	384	Matrigel	-	CellTiterGlo (viability)	[83]
Colorectal cancer organoids	>500	384	BME	Bispecific antibodies	Imaging-based	[84]
Colorectal cancer	Multi-dose tests of compounds alone or in combination, 11 patient-derived organoid types as single-point or multiple biological replications	384	BME	Targeted inhibitors and combinations	CellTiterGlo 2.0 (viability)	[85]
Colorectal cancer	56 compounds, 20 cancer and 6 normal colonic organoid lines	384	Matrigel	Chemotherapeutics/Targeted	Imaging-based	[86]
Colorectal and gastroesophageal cancers	55 compounds, 19 organoid types, triplicate	96	Matrigel	FDA-approved	CellTiterBlue (viability)	[67]
Gastric cancer	37 compounds, 9 organoid types (from 7 patients), 7 concentrations in triplicate with 2 biological replicates	384	Matrigel	FDA-approved	CellTiterGlo 2.0 (viability)	[11]
Liver cancer	29 compounds in biological and technical duplicate, 6 organoid lines, 7 concentrations	384	BME2	Mixed FDA-approved/targeted	CellTiterGlo (viability)	[87]
Pancreatic cancer	1172 compounds, 2 organoid types	384	Matrigel	FDA-approved	CellTiterGlo 3D (viability)	[66]
Pancreatic cancer	5 compounds, multi-dose in singlet or triplicate	-	Matrigel	Chemotherapeutics	CellTiterGlo (viability)	[88]
Ovarian cancer organoids	23 compounds at 6 concentrations in technical triplicate	48	Matrigel	FDA-approved	CellTiterGlo-3D (viability)	[81]
Bladder cancer	50 compounds on 9 organoid lines, triplicate at 5–7 concentrations	96	ULA	FDA-approved/kinase inhibitors	CellTiterGlo (viability)	[89]
Renal cell carcinoma	5 compounds, 8 concentrations, biological triplicate, 4 organoid lines	96	TC-treated (2D)	Tyrosine kinase inhibitors	CellTiter 96^®^ AQueous One (proliferation)	[90]
Castration-resistant prostate cancer	126 compounds at 6 doses, 4 organoids	384	TC-treated (2D)	Chemotherapeutics/targeted	CellTiterGlo (viability)	[91]
Lung cancer	3 compounds, 10 concentrations, triplicate	96	Matrigel	FDA-approved drugs	CellTiterGlo (viability)	[92]
Malignant rhabdoid tumor organoids	146 compounds at 6 concentrations in 6 organoid types	384	ULA	Mixed	CellTiterGlo-3D (viability)	[93]
Breast cancer	6 compounds in 28 organoid lines, 21 concentrations	384	BME	Targeted inhibitors	CellTiterGlo 3D (viability)	[12]

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
