# Peer review of "Tumor Organoids as a Research Tool: How to Exploit Them"

_cells, 2022, doi:10.3390/cells11213440_

Round 1
Reviewer 1 Report
The work presented by Booij and others seems to be interesting. It provides a summary of very important research on organoids. The authors refer to a wide range of literature and the examples they cite are present. I recommend the work for publication.
Author Response
Thank you for your time and kind review of our work.
Reviewer 2 Report
This is a very interesting article. The manuscript is well-written and clear to the readers.
Specific comments
Where is the perspective I would wish from a review?
I would like to see a more detailed conclusion at the end of the article.
Author Response
Thank you for your time and comments. In the revised manuscript we added a paragraph on perspectives of organoid cultures and gave some more details in the conclusions [lines 531 to 546].
Reviewer 3 Report
Tijmen and co-workers summarize the recent development of organoid for cancer research. And discuss the model, application and testing method. Organoids is a collection of organ-specific cell types that self-organizes in a 3D structure recapitulating the process of self-organization during development in vivo and with functions similar to the organ. It’s important and help researchers to understand the process of the development of cancer. However, in this review, the title does not accurate the context that the author summarizes in main text and there are many other places need to be edited and expanded.
The specific comments as follow:
1, not the only reason that minority of highly aggressive tumor cells are able to be cultured classically in vitro as monolayers. There are many other important reasons, the author should expand the reason why researcher’s consider organoid cultures in the paragraph Line 30-line 33.
2, “one can use nanofibrillar cellulose or alginate gels, or use synthetic polymers or recombinant peptides. Synthetic hydrogels, as well as nanocellulose or alginate gels” The author should cite the references on each example.
3, “These materials can also be used in combination with other materials, such as combination of collagen and nanocellulose” The author should indicate what’s “These materials” representative”. And should show why they combine or what’s the advantage if combination with collagen or nanocellulose.
4, “but also collagen suffers from the same temperature-dependent polymerization that makes Matrigel less ideal for lab automation purposes.” it makes sense to change Matrigel to collagen in this sentence. The author should double check it.
5, In the subtitle of “Improved Substrated and Cultures” The author should delete the sentence of “It is not the scope of this review to 83 elaborate on the varieties of matrices that have been used for organoid cultures, although 84 we gladly provide some suggestions for further reading on this topic [24–29]”
6, The author should cite the references here after this sentences:“One possible way to study tumor cells in the context of its own TME has been proposed with the advent of Air-Liquid Interface (ALI) culture” like the representative references ( cited 35, Neal JT, et al. Organoid modeling of the tumor immune microenvironment. Cell. 2018;175(7):1972–1988.e16.) and Finnberg NK, et al. Application of 3D tumoroid systems to define immune and cytotoxic therapeutic responses based on tumoroid and tissue slice culture molecular signatures. Oncotarget. 2017;8(40): 66747 –57.
7, Delete the sentence of “In the next section 112 we will focus on the available alternative methods to study the interaction between tumor 113 and immune cells, exploiting the organoids technology”.
8: Cite the refence 42 after sentence “Tsai et al. describe how to co-culture human pancreatic cancer organ- 134 oid together with autologous CAFs and T-cells.” As well.
9, The author using subtitle “3. Advanced Testing”. It seems application was more accurate. and author should summarize more advanced application of organoids and put the some of the context for advanced testing in Advanced Readout.
10, The author should edit the title for accurate describe what’s this review included.
Author Response
Thank you for your time and comments. Indeed, our initial working title with “tumor organoids as research tool: why, when and how to exploit them”, may not accurately describe our review which is more focused on the “how”. We therefore adapted the title accordingly.
In regard to the raised comments:
- We agree that there are other reasons that organoid models should be considered and expanded this in [lines 33 to 36].
- References to these statements have been added in the revised text. Considering that the research body on synthetic polymers and recombinant peptides in organoid research and 3D cell culture assays is rather large, we cited informative reviews that evaluate these technologies. [lines 83 to 84].
- “These materials” have been specified and the advantage of a combination with collagen and nanocellulose described by including the following statement “which can greatly reduce the amount of reagents required, and also allow for faster polymerization” [lines 81 to 82].
- Both collagen and Matrigel have the same temperature-dependent polymerization issues. We re-formulated the statement [lines 78 to 79].
- Sentence has been deleted as suggested but references were retained [line 88].
- The two suggested references have been added [line 118].
- The sentence has been deleted as suggested [line 124].
- The reference has been added right after the sentence [line 145].
- The title has been changed according to the reviewers’ suggestions into applications [line 256]
- Title has been shortened to better describe content of review.
Round 2
Reviewer 3 Report
I have no comments now.